# Super-enhancer-associated gene *CAPG* promotes AML progression

Qian Ma[1], Minyi Zhao[2], Bing Long[3] & Haixia Li [1✉]

Acute myeloid leukemia is the most common acute leukemia in adults, the barrier of refractory and drug resistance has yet to be conquered in the clinical. Abnormal gene expression and epigenetic changes play an important role in pathogenesis and treatment. A super-enhancer is an epigenetic modifier that promotes pro-tumor genes and drug resistance by activating oncogene transcription. Multi-omics integrative analysis identifies the super-enhancer-associated gene *CAPG* and its high expression level was correlated with poor prognosis in AML. CAPG is a cytoskeleton protein but has an unclear function in AML. Here we show the molecular function of *CAPG* in regulating NF-κB signaling pathway by proteomic and epigenomic analysis. Knockdown of *Capg* in the AML murine model resulted in exhausted AML cells and prolonged survival of AML mice. In conclusion, SEs-associated gene CAPG can contributes to AML progression through NF-κB.

[1] Department of Clinical Laboratory, Peking University First Hospital, Beijing, China. [2] Department of Hematology, The Seventh Affiliated Hospital, Sun Yat-sen University, Shenzhen, China. [3] Department of Hematology, The Third Affiliated Hospital, Sun Yat-sen University, Guangzhou, China.
✉email: bdyylhx@126.com

Acute myeloid leukemia (AML) is an aggressive hematologic malignancy that is caused mainly by the accumulation of genetic mutations and malignant proliferation of hematopoietic stem cells (HSCs)[1–5]. The high rates of drug resistance and recurrence are the current challenge in clinical trials[6]. AML can occur and progress as a result of random genetic mutations and epigenetic alterations[7–10].

Recent years have highlighted significant progress in understanding the underlying abnormal epigenetic patterns of several cancers. Epigenetic abnormalities activate proto-oncogenes[11] and cause chromosomal instability[12,13], while transcriptional silence of tumor suppressor genes[14,15]. Detection of epigenetic molecular events during the development and progression of AML could be valuable for the development of more effective treatments.

Super-enhancer (SE) is essentially a cluster of active typical enhancers (TE)[16]. Compared with TEs, SEs have a larger size and an increased capability to activate transcription. During development, SEs play critical roles in regulating cell fate and determining genes[16]. Cancer cells acquire SEs at oncogene regions such as *MYC*, *BCL2*, and their cancerous phenotype relies on abnormal transcription[16,17]. Epigenetic modifications such as DNA methylation, histone acetylation can regulate downstream effectors' abnormal activation through SEs[18–20]. SEs were also found to mediate tumor-related genes[21,22], immune checkpoint[23], inflammatory cytokines[24] transcription[25,26]. These suggest that SEs are involved in the occurrence and development of tumor and drug resistance recurrence.

Capping actin protein, gelsolin-like (CAPG) is an actin-binding protein of the gelsolin superfamily that serves a crucial role in the organization of the actin cytoskeleton[27]. Evidence suggests that proteins belonging to the gelsolin superfamily may be involved in other processes, including gene expression regulation[28,29]. Unlike other gelsolin superfamily members, CAPG is mostly found in the nucleus rather than cytoplasm[30]. Previous studies have demonstrated that CAPG is a breast cancer biomarker for bone metastasis development and treatment[31]. However, the expression pattern and function of CAPG in AML remain to be investigated.

In this study, we identified an AML-specific super-enhancer-associated gene *CAPG*. and verified that *CAPG* can regulate the progression of AML through NF-κB signaling pathway.

## Result

***CAPG* is an AML-specific super-enhancer-associated gene**. In-depth understanding the pathogenesis of AML, conferred a new treatment target for AML therapy, we first identified AML-specific super-enhancers and SEs-associated genes. Super-enhancers are important for controlling and defining the expression of cell-specific genes[16]. To identify AML-specific SEs and related genes, we integrated multi-omics data to characterize the AML-specific super-enhancer-associated gene with the aim of providing new targets for AML therapy. We identified acute myeloid leukemia-specific SEs and SEs-associated genes by analyzing AML H3K27ac chromatin immunoprecipitation sequencing (ChIP-seq) data. Meanwhile, SEs associate genes of normal blood cells, including neutrophils (NEs), monocytes (MOs), and hematopoietic stem cell progenitor cells (HSPCs) were identified (Supplementary Fig. 1). Furthermore, we analyzed publicly available RNA-seq dataset (GSE128910) from previous study, calculated differentially expressed genes between health volunteers and AML patients, picked out the genes that were expressed in AML (RPKM > 1) and were 3 times higher than normal (Fig. 1a)[32–35].

According to the above analysis, we screened out 6 AML-specific SEs-associated genes (*CAPG*, *CD207*, *GPR132*, *SLC7A11*,

*HIPK3,* and *FCER1G*) that have a significantly high expression (Fig. 1b). These gene regions are enriched for SEs and highly expressed in AML (Fig. 2a, Supplementary Fig. 2a). We selected the *CAPG* as the object of our research, which has been reported to high expression in AML.

***CAPG* is highly expressed in MLL-AF9-induced murine AML cells**. We analyzed The Cancer Genome Atlas (TCGA) database and found that *CAPG* expression level was positively associated with poor prognosis in AML (Fig. 2b). In parallel, the expression of other SEs-associated genes was also correlated with prognosis in varying degrees (Supplementary Fig. 3). In addition, an investigation by TCGA database revealed that *CAPG* was statistically significantly highly expressed in a variety of tumor tissues compared to normal tissues (Fig. 2c). This implies that *CAPG* is closely related with tumor development.

To further validate whether SEs-associated genes are differentially expressed in the disease and affect the AML prognosis, the expression levels in the collected peripheral blood samples of patients and bone marrow samples of mouse models were subsequently compared (Fig. 2d). We observed that *CAPG* expression in MLL-AF9-induced murine AML cells was significantly elevated compared to the normal murine bone marrow cells (Fig. 2e, f and Supplementary Fig. 2b).

In general, the oncogenes in AML can be accurately screened by super-enhancers-associated genes, and the super-enhancer-associated gene *CAPG* is related to the progression of AML.

**Identification of CAPG interactomes**. CAPG is known as a member of the gelsolin superfamily of proteins, which regulates actin filament length by capping or severing filaments[36]. CAPG is located in the cellular cytoplasm and nucleus, unlike other members of this family[37]. *CAPG* has been linked to a poor prognosis in pancreatic[38] and breast Cancer[31,39], but no research in AML has been conducted. In AML, there is an upregulation trend in the expression of actin (Supplementary Fig. 2c). To examine how *CAPG* contributes to AML progression, we purified and characterized CAPG protein complexes in the human AML cell line THP-1 to construct CAPG *interactomes by immuno-precipitation with mass spectrometry (IP-MS) (Fig. 3a and Supplementary Data 3).

We identified 79 CAPG-interacting proteins in THP-1 (Supplementary Material). Moreover, enrichment analysis via gene ontology and conclusion with graph-based visualization of interactomes were employed for cellular biological functions determined. Gene Ontology was utilized for selecting the functional annotation clusters. The functional analysis revealed that, as an actin-binding protein, CAPG protein partners from total protein extracts are intimately associated with nuclear acid and actin filament binding and are also involved in cellular molecular activities. GO categories were involved in processes related to cellular metabolism, localization, and stability of biomolecules. It is also involved with RNA splicing and can influence gene expression, engaged in epigenetic control, related to chromatin remodeling, and histone modification. CAPG was found in the ribosome, nuclear, and nucleoplasm, particularly in the actin cytoskeleton, according to cellular component analysis. CAPG also associated with epigenetic modification complex. The result indicated that CAPG as a gelsolin protein not only constitutes the cytoskeleton, but also plays a critical function in cellular growth processes (Supplementary Fig. 4a).

**CAPG links to NF-κB signaling pathway in AML**. To further interpret the potential functional role of *CAPG*, interactome results were subjected to protein-protein interactions (PPI)

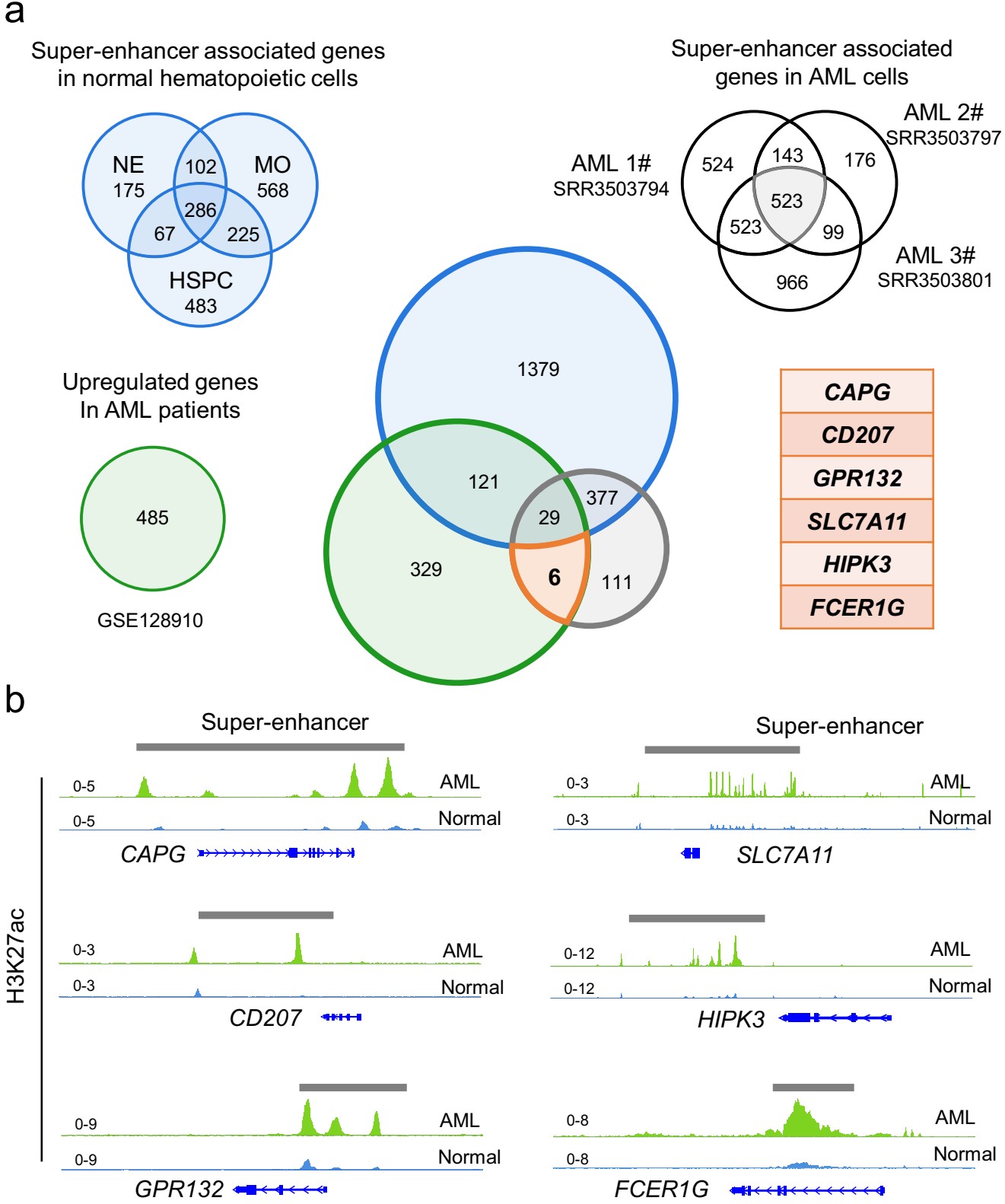

**Fig. 1 Identification of super-enhancers and associated genes. a** Experimental scheme to search for specific and highly expressed super-enhancer-associated genes in AML cells. Neutrophils (NEs), monocytes (MOs), and hematopoietic stem cell progenitor cells (HSPCs) represent normal blood cells. Blue circle represent Super-enhancer associated genes in normal hematopoietic cells. Green circle represents upregulated genes in AML patients. Gray circle represents super-enhancer-associated genes in AML cells. **b** The ChIP-seq tracks show the representative H3K27ac signal in healthy volunteers and AML patients. The super-enhancers are shown as gray boxes.

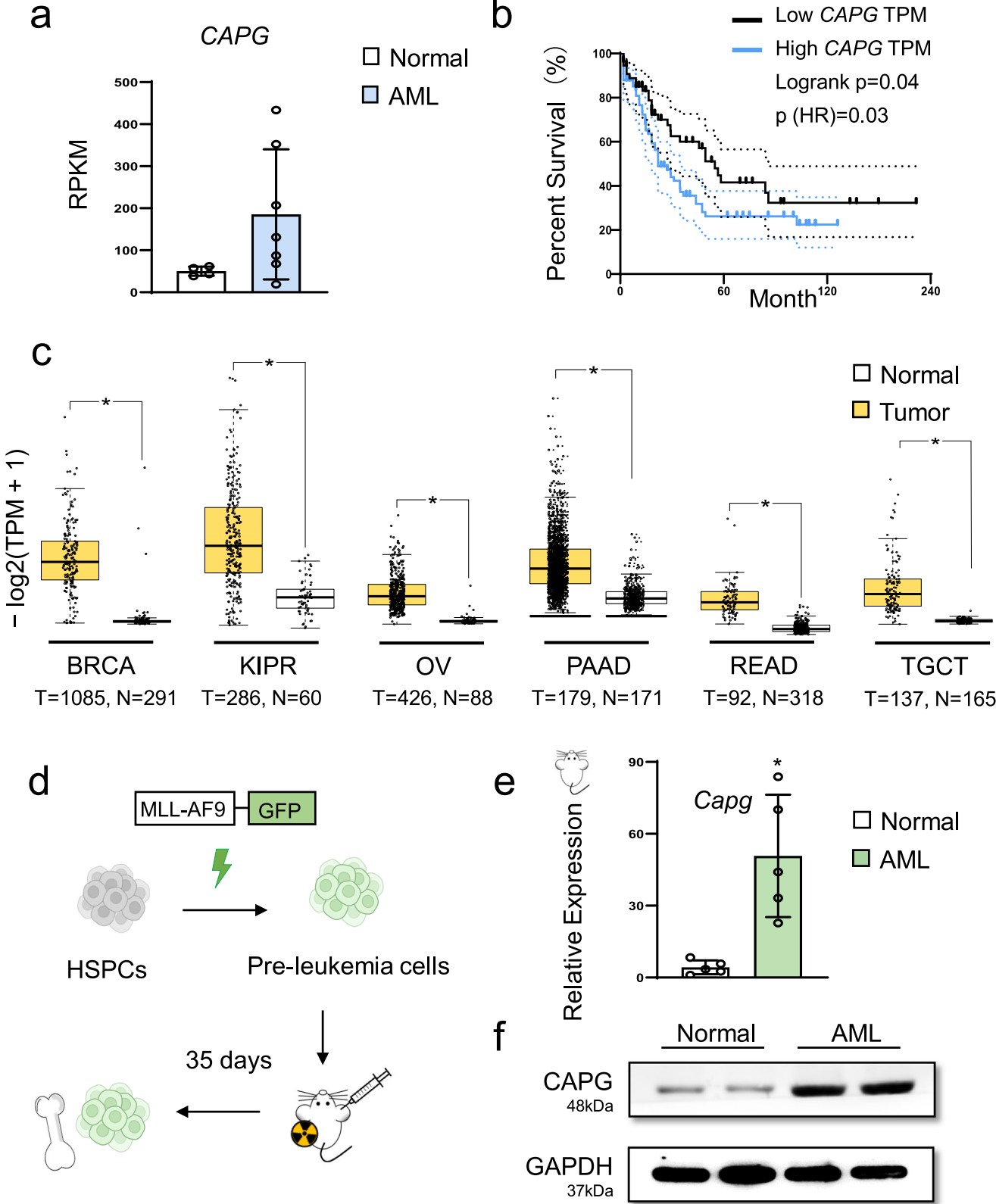

network analysis (Fig. 3b). Notably, we observed that CAPG physically interacts with multiple cytoskeletal protein complexes involving myosin complex, troponin complex, tubulin complex, and actin complex. This result is consistent with the function of CAPG as a gelsolin protein.

Furthermore, analyzing the characteristics of the interactome revealed that CAPG is associated with multiple epigenetic regulatory complexes, which have not been reported previously. Of note, three interacting proteins (CCAR2, RPL4, ZFP91) have been reported as the nuclear factor-κB (NF-κB) signaling pathway activators[40–42] (Fig. 3c). In addition, SNW1 complex has been identified as a novel transcriptional regulator of the NF-κB pathway[43] (Supplementary Fig. 4b). Researches show that NF-κB activity imbalance causes inflammation-related diseases such as

**Fig. 2 *CAPG* is an AML-specific super-enhancer-associated gene. a** RNA-seq data (GSE128910) shows *CAPG* is highly expressed in AML patients. Healthy volunteers (*n* = 4) or AML patients (*n* = 7). **b** The Kaplan–Meier survival curves of *CAPG* in The Cancer Genome Atlas (TCGA)-LAML database. Dot lines represent 95% Confidence bound. **c** Compare the expression level of *CAPG* in breast invasive carcinoma (BRCA), kidney renal papillary cell carcinoma (KIPR), ovarian serous cystadenocarcinoma (OV), pancreatic adenocarcinoma (PAAD), rectum adenocarcinoma (READ) and testicular germ cell tumors (TGCT) from tumor and normal tissue. Wilcoxon Rank Sum and Signed Rank Test were used to analyze the significance of differences. The median is shown by the line inside the box. **d** Experimental scheme for MLL-AF9-induced AML mice established. **e** RT-qPCR showing that *CAPG* expression was increased in murine leukemic cells from primary transplant mice compared with normal mice bone marrow cells (*n* = 5 mice). Each dot represents a mouse. Data are presented as means ± SD. \**p* < 0.05 compared with control group. **f** Western blot showing that CAPG expression was increased in leukemic cells from primary transplant mice compared with normal mouse bone marrow cells. GAPDH was used to show equal loading. Each band represents an individual mouse.

cancers, so that NF-κB is considered a potential target for cancer therapy[44]. Activation of NF-κB pathways contributes to the leukemic transformation initiated by some crucial oncogenic kinases[45]. We detected the characterized CAPG interacts with NF-κB signaling pathway complex in AML cells, so we hypothesized that *CAPG* has potential as a target to regulate the disease process of AML.

**CAPG regulates AML progression through NF-κB signaling pathway**. To elucidate the connection between *CAPG* and NF-κB during AML development, we collected bone marrow cells from normal and AML mice. We performed CAPG ChIP-seq assays to identify the gene regulatory network and evaluated the data to select AML-specific peaks enriched in gene regions (Fig. 4a).

GO analysis showed that *CAPG*-regulated genes were substantially related to the NF-κB pathway in terms of cellular component and molecular function (Fig. 4b). We also assessed transcription factor targets and downstream genes, identified *CAPG* as a potential gene involved in regulating the transcriptional activity of the NF-κB pathway (Supplementary Fig. 5a, b). All this evidence implies that *CAPG* is related to the activation of the NF-κB pathway.

To further demonstrate our conclusion, we investigated the enrichment of *CAPG* in the genome of THP-1 cells. ChIP-qPCR analysis showed that CAPG was significantly enriched in the transcription factors region of NF-κB in THP-1 cells (Supplementary Fig. 5c).

Additionally, we downregulated *CAPG* expression in THP-1 cells (Fig. 4c and Supplementary Fig. 5d)) and assessed the expression levels of downstream genes of NF-κB (Fig. 4d). Notably, *CAPG* knockdown led to a marked reduction in the expression of several apoptotic and immune response genes that have been associated with AML progression (Fig. 4d)[46,47]. We also collected RNA-seq data after knocking down *CAPG* and performed KEGG analysis, which revealed that the downregulated genes were significantly enriched in the NF-κB pathway (Fig. 4e and Supplementary Data 4).

To demonstrate the direct regulation of NF-κB pathway by CAPG, we activated the pathway by stimulating THP-1 cells with TNF-α and observed a significant rescue of downstream genes expression (Fig. 4d). Moreover, when *CAPG* is depleted, the expression level of leukemogenesis is significantly reduced (Fig. 4f). These observations imply that CAPG can directly modulate the NF-κB pathway, which in turn affects the progression of AML.

**Capg knockdown inhibits AML progression in vivo**. To determine whether *CAPG* is associated with AML progression, we conducted *CAPG* knockdown experiments to verify its function. Knockdown of *CAPG* can inhibit the growth of AML cells (Supplementary Fig. 6a). To further explore the potential pro-tumor role of *CAPG* in AML, equal numbers of control (Ctrl) or

*Capg* knockdown (sh*Capg*) murine AML cells were transplanted into syngeneic wild-type (WT) recipients (Fig. 5a).

To assess the effects of *Capg* reduction, we first sorted green fluorescent protein (GFP)+ leukemia cells from Vector and sh*Capg* AML mice and proved that the expression levels of sh*Capg* group were reduced by RT-qPCR and western blot (Fig. 5b, c).

We found that deficiency of *Capg* significantly exhausted AML cells in the peripheral blood (PB) (Fig. 5e) and reduced disease burden in the bone marrow (Fig. 5d).

Furthermore, the spleen and liver of AML mice were significantly enlarged, whereas *Capg* knockdown significantly relieved these symptoms (Fig. 5f). Consistently, histological analysis showed that AML mice in sh*Capg* group had fewer leukemia cell infiltration in peripheral blood (Fig. 5g). Importantly, *Capg* knockdown significantly prolonged the survival of AML mice, median overall survival (MOS) expended to 83 days in sh*Capg* group compared to 51 days in the control group (Fig. 5h). These results indicate that *Capg* knockdown suppresses the progression of MLL-AF9-induced AML in mice, which supports our hypothesis that *Capg* is oncogenic in AML.

Further, summarize the pattern, *CAPG* acts on the NF-κB signaling pathway by protein interactomes to activate the expression of downstream genes. It can accelerate the progression of AML by directly binding to NF-κB family transcription factors regions and turning on the regulation of downstream gene expression.

Taken together, the data shows that *CAPG* plays an important role in AML progression by regulating the NF-κB signaling pathway. Our study identified a super-enhancer-associated gene *CAPG* as an oncogene in AML, which conferred a new treatment target for AML therapy.

## Discussion
Epigenetic modifications often play a key role in the occurrence and development of various tumors, and super-enhancers (SEs) are an epigenetic element that can regulate cell-type-specific gene expression and play a crucial role in cell fate decisions, including the transcription of tumor cell immune checkpoint molecules[23], immune escape[25,26], and the expression of inflammatory cytokines[24]. RNA-seq analysis at the transcriptome level can generate a large number of differentially expressed genes, making it difficult to identify genes that significantly impact disease progression. In contrast, super-enhancers are highly cell-type specific and have powerful gene expression regulatory functions, which can drive the expression of genes that control and define cell identity[16]. By integrating super-enhancer data with transcriptomic data, we can narrow the target range and improve accuracy. It remains to be verified and explored whether combining cell-type-specific SEs and transcriptomics can achieve a more accurate and efficient screening of pro-tumor genes.

In this study, we identified SE-associated genes by integrating cell-specific SEs with RNA-seq data and found six genes (*CAPG*,

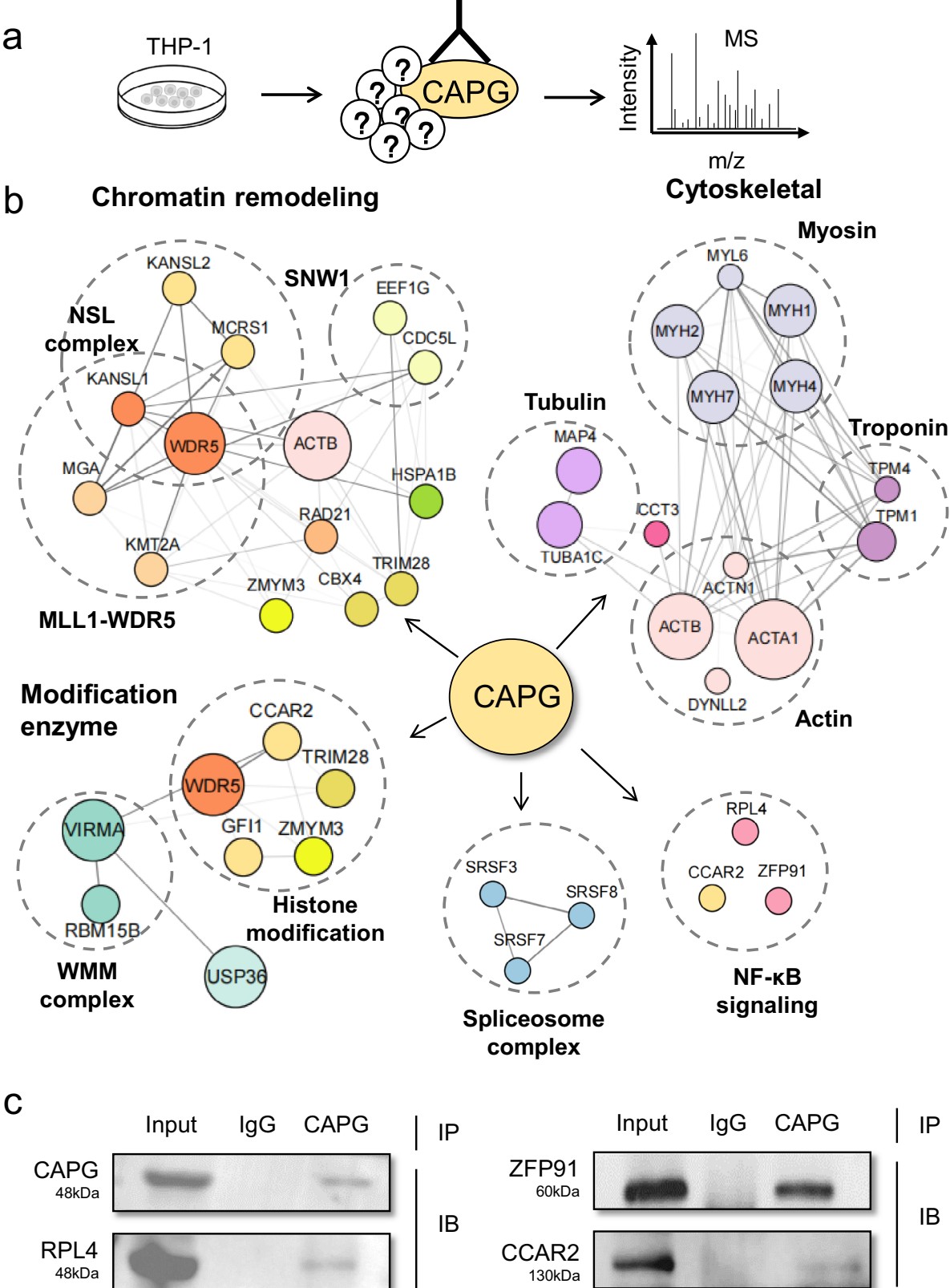

**Fig. 3 CAPG links to NF-κB signaling pathway in AML. a** Scheme showing the antibody-dependent IP-MS system for identifying CAPG interactomes in THP-1 cell. **b** Connection of CAPG with multiple protein complexes. Summary of major protein complexes (dashed line circle) associated with the CAPG interactome. The size of the circle represents the reliability of protein-protein interaction, and the thickness of the connecting line represents the interaction intensity. **c** Co-IP shows the interactions between CAPG and NF-κB related proteins (RPL4, ZFP91, CCAR2).

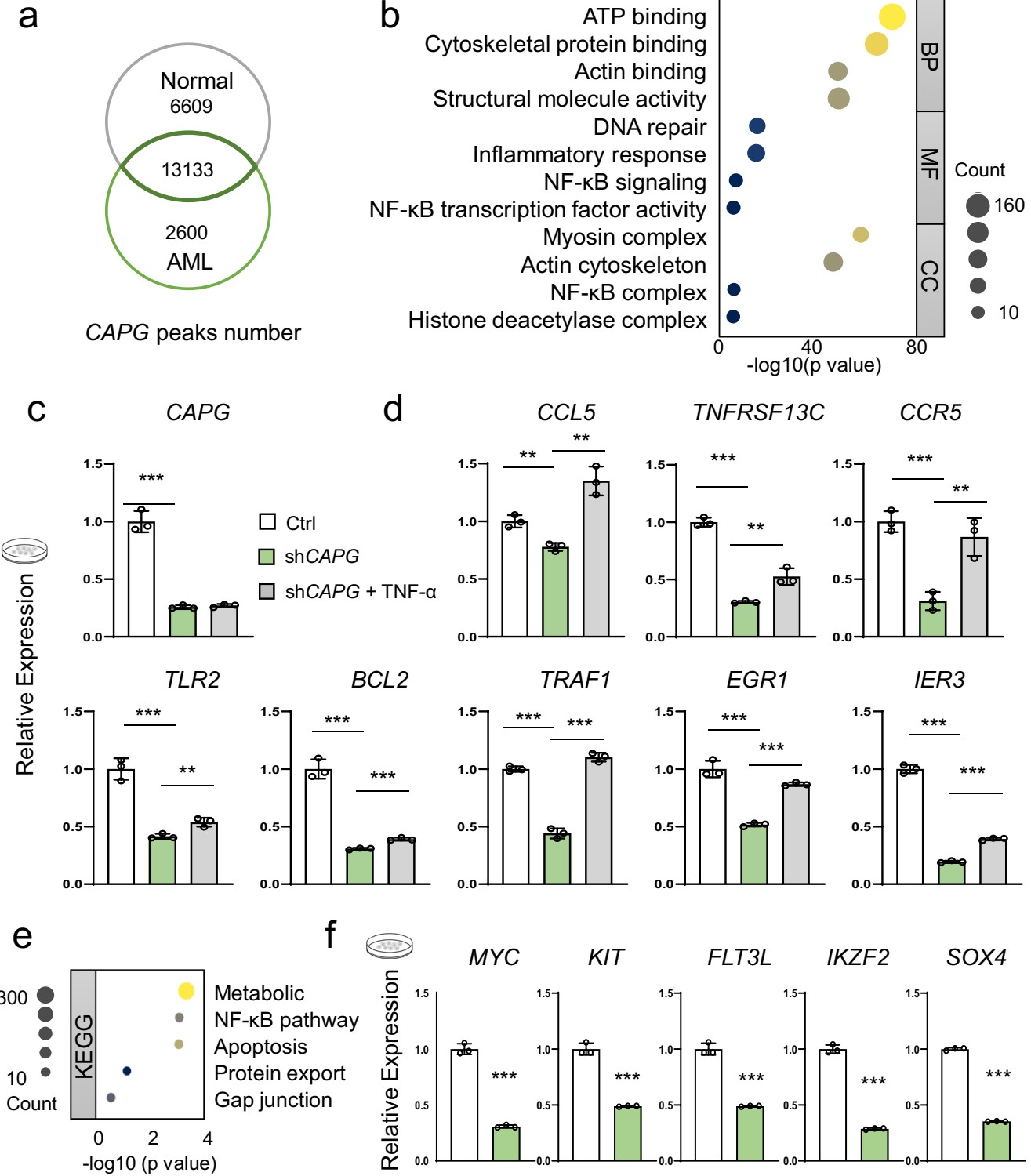

**Fig. 4 *CAPG* regulates AML progression through NF-κB signaling pathway. a** Venn diagram of the *CAPG* binding loci in normal bone marrow cells and AML bone marrow cells. **b** Gene Ontology term enrichment analysis of *CAPG*-regulated genes only in AML cells, *P* < 0.05 for each result displayed on the figure. **c** mRNA levels of *CAPG* in control, knockdown, and 10 ng/mL TNF-α stimulated group were compared. Data are presented as means ± SD. ***p* < 0.001. The results are from three biological replicates. **d** The addition of TNF-α reversed the inhibition of NF-κB pathway activity induced by *CAPG* knockdown in THP-1 cells. Data are presented as means ± SD. ***p* < 0.01, ****p* < 0.001. Results are from three biological replicates. **e** The KEGG analysis revealed that *CAPG* downstream genes were mainly enriched in the NF-κB signaling pathway, *P* < 0.05 for each result displayed in the figure. **f** mRNA levels of leukemogenesis in control, CAPG knockdown group. Data are presented as means ± SD. ***p* < 0.01, ****p* < 0.001.

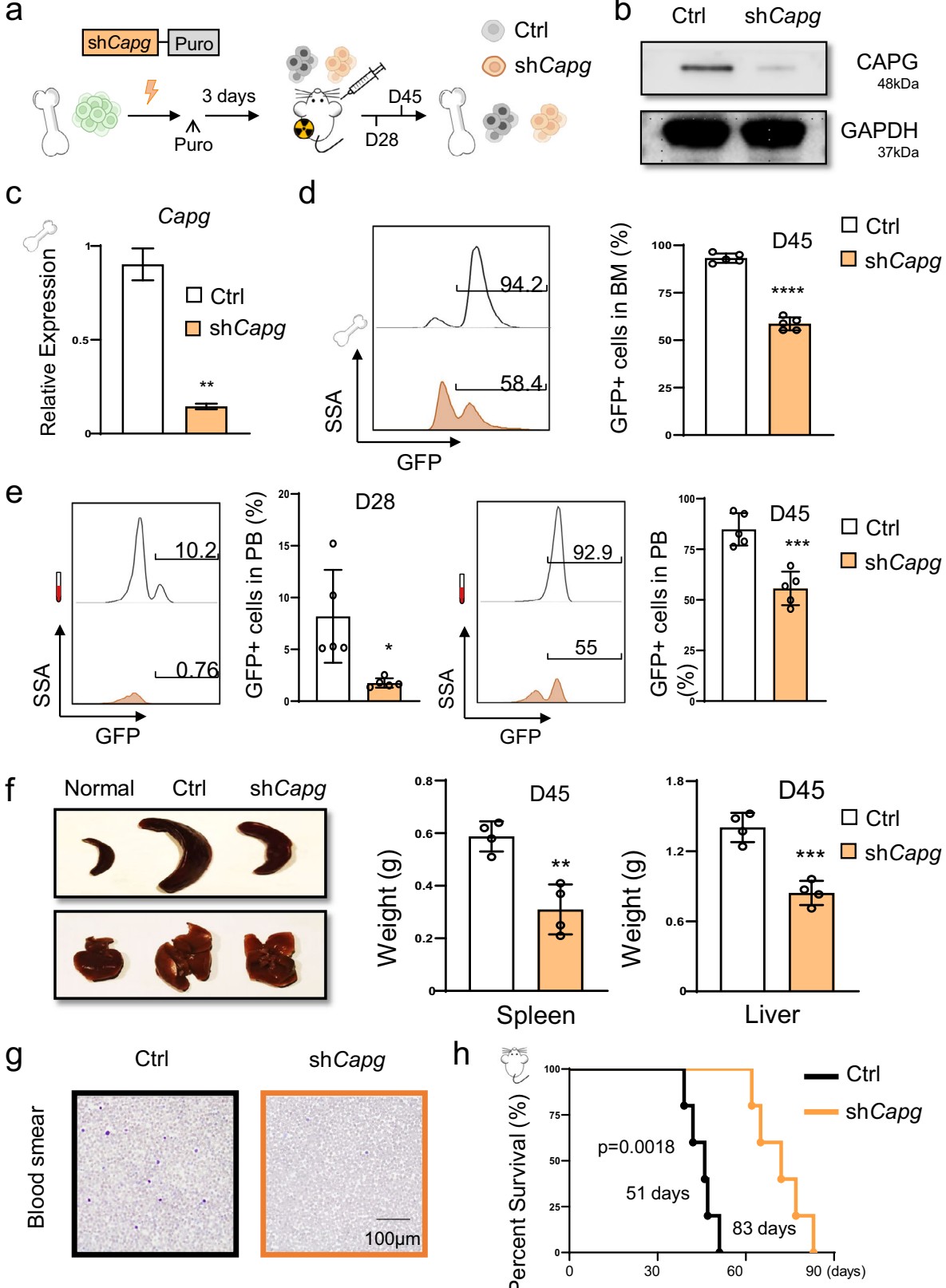

CD207, GPR132, SLC7A11, HIPK3 and FCER1G) that are correlated with poor prognosis in acute myeloid leukemia (AML) patients according to the TCGA-LAML database. *CD207* is a glycosylated receptor that acts as a specific receptor of Langerhans cells, processing of antigen for presentation to T cells[48]. *GPR132* is a leukemia orphan receptor that has the potential to trigger myeloid differentiation[49]. In acute myeloid leukemia patients, high expression of *SLC7A11* is associated with poor prognosis[50,51]. *HIPK3* is involved in the apoptosis process and is associated with poor prognosis in various types of cancer[52]. *FCER1G*, as a component of the high-affinity immunoglobulin E (IgE) receptor, is associated with the immune response in various

**Fig. 5 *Capg* knockdown inhibits AML progression in vivo. a** Experimental scheme for *Capg* knockdown in vivo. **b** Western blot analysis showing *Capg* knockdown in sorted AML cells from AML mice bone marrow. **c** RT-qPCR analysis showing *Capg* knockdown in sorted AML cells from bone marrow of scramble control (Ctrl) and *Capg* knockdown (sh*Capg*) AML mice at day 45 post-transplantation. Data are presented as means ± SD. ***p* < 0.01 compared with the control group. The results are from three biological replicates. **d** Representative cytometric flow plots, the percentage of GFP + AML cells in bone marrow (BM) at day 45 post-transplantation ($n = 5$ mice). Each dot represents a mouse. ****p* < 0.001 compared with control group. **e** Representative cytometric flow plots and statistic results show that *Capg* knockdown decreases leukemia burden in peripheral blood (PB) at day 28 and day 45 post-transplantation ($n = 5$ mice). Each dot represents a mouse. **p* < 0.05, ****p* < 0.001 compared with control group. **f** Representative image of the spleen (upper left), liver (bottom left), and quantitative analysis of spleen weight (middle) and liver weight (right) from scramble control and *Capg* knockdown AML mice ($n = 5$ mice). Each dot represents a mouse. ***p* < 0.01, ****p* < 0.001 compared with control group. **g** Wright–Giemsa staining of a blood smear from control and *Capg* knockdown AML mice. Scale bar 20 µm. **h** Survival analysis of mice transplanted with scramble control or *Capg* knockdown AML cells. Data shown are combined from two independent transplants ($n = 5$ mice). $p = 0.0018$, log-rank test.

types of cancer[53,54]. *CAPG* has been reported to be closely associated with protein-tyrosine kinases in AML, linked to drug resistance in ALL[55,56], and is associated with poor prognosis in pancreatic[38] and breast Cancer[31,39]. *CAPG* has been reported to be closely associated with protein-tyrosine kinases in AML and linked to drug resistance in ALL[55,56]. Next, we verified the function of *Capg* in AML in a mouse model and found that reduced *Capg* expression can hinder the progression of AML disease. Taken together, these data suggest super-enhancer-associated gene *CAPG* is a potential therapeutic target for AML, and the reliability of the integrate analysis approach based on super-enhancer, also a promising tool for predicting biomarkers of diseases.

Super-enhancers play an important role in tumorigenesis and tumor progression. In this study, we found the related gene *CAPG* through the super-enhancer, and verified its role in AML. The fundamental biological mechanism is the promotion of AML through the activation of downstream leukemogenesis expression through epigenetic element SE, is a classic event of cell fate transition through dynamic epigenetic alterations. Our study demonstrates at the genomic level that *CAPG* knockdown impeded AML disease progression. Gene editing technology permits precise molecular alterations at the genome level for the treatment of many diseases[57]. In recent years, the application of CRISPR technology to epigenetic molecular modification has been fruitful[58]. The use of CRISPR-dCas9 combined with histone deacetylase to erase histone modifications and decrease SE activity will have the potential to restrict SE-related gene expression and hamper the disease progression[59]. We speculate that at the epigenetic level, the proto-oncogene *CAPG* expression will be reduced by inhibitory activity of SE, which will limiting the progression of AML.

Genomic instability and epigenetic abnormality can directly facilitate disease occurrence. Epigenetic alterations, including DNA methylation, histone modification, chromatin accessibility have been reported to play a major role in gene expression regulation. According to "genetic central dogma" of molecular biology, genetic information is transmitted by way of DNA-RNA-protein. As the last link of this process, proteins function as the main undertaker of vital movement by forming macromolecular complexes. There is increasing evidence depicting the essentiality of protein-protein interactions (PPIs) governing a wide array of cellular processes.

*CAPG* was reported to be an actin-binding protein of the gelsolin superfamily, associated with the cytoskeleton organization. Increased *CAPG* expression has been found in several metastatic cancers, suggesting its role in cancer cell invasion and metastasis[60–62]. Therefore, we identified the CAPG interactomes by immunoprecipitation with mass spectrometry (IP-MS). We uncovered in the CAPG interactome components of regulatory protein complexes such as chromatin-remodeling protein NSL complex, SNW1 complex, MLL1-WDR5 complex,

and modification enzyme protein involving WMM complex. Multiple independent studies have found a physical link between cancer aggressiveness and these epigenetic regulatory complexes. It is well established that NSL complex is correlate with cancer aggressiveness[63] and poor survival[64,65]. Moreover, disrupting the MLL1-WDR5 interaction is considered a therapeutic approach for leukemia[66]. Indicating the importance of these epigenetic regulatory pathways and factors for AML progression. Next, we analyzed the molecular function of the interaction partners and found three of them (*CCAR2, RPL4, ZFP91*) have been reported as the activator of NF-κB signaling pathway. *CCAR2*, as known as *DBC1*, can regulate anoikis through NF-κB pathway[40,67]. *CCAR2* stimulates the phosphorylation of nuclear relA (*RelA*), enhancing the transcriptional activity of NF-κB and up-regulating target genes which are associated with anoikis resistance[40,67]. Meanwhile, *RPL4* and *ZFP91* as the activators of NF-κB pathway, induce the proliferation and promote the process of cancer[41,68,69]. ZFP91 promotes cancer cell proliferation and carcinogenesis by activating the transcriptional coregulatory protein HIF-1 via NF-κB[68]. *RPL4* involved in a ribosomal protein complex that activates NF-κB via CD40[41]. SWNI is likewise linked to the NF-κB pathway in the CAPG interactome, its involvement entails binding to the NF-κB to facilitate transcriptional elongation of target genes[43]. Above results indicate that *CAPG* participates in regulating NF-κB signal transductions which have not been reported before.

Furthermore, we utilized CAPG ChIP-seq data to confirm the aforementioned findings. In addition to the well-known roles in molecular binding and cytoskeleton structure, GO and motif analysis showed that CAPG-regulated genes were strongly linked to the NF-κB pathway. *CAPG* knockdown significantly reduced NF-κB downstream genes expression. All of this support the assertion made above that CAPG is involved in the regulation of the NF-κB signaling pathway. This indicates that the identification of protein interactomes can provide a basis for studying protein functions, and is an important way and future trend to study the potential functions of encoding genes.

Taken together, the expression of super-enhancer-associated gene *CAPG* corresponds with progression in AML and is connected with NF-κB pathway activation. We further postulated that *CAPG* could be a viable therapeutic target for AML, and the reliability of the comprehensive analytic algorithm relies on super-enhancer, also a potential tool for predicting biomarkers of pathologies.

## Methods

**Cell culture**. The human monocytic cell line THP-1 cells were purchased from Solarbio (China), were cultured in RPMI media (Corning, 10-040-CV) supplemented with 10% FBS and 1% penicillin and streptomycin and regularly tested negative for mycoplasma contamination using PCR. Cells were stimulated with TNF-α (10 ng/ml) in re-activating the NF-κB pathway experiment.

**Definition of super-enhancers and SE-associated genes**. To identify typical enhancers, we initially stitched the public H3K27ac ChIP-seq database (NE SRR1915572, MO SRR787551, HSPC SRR2094192, AML1# SRR3503794, AML2# SRR3503797, AML2# SRR3503801) peaks along 500 bp away, but re-split them if the gaps coverage is above 0.4. Then the stitched raw enhancer together with aligned H3K27ac and input reads were used to run the ROSE algorithm. Briefly, constituent enhancers were stitched together if they are within a certain distance and ranked by their input-subtracted signal of H3K27ac. And then, we separated super-enhancers from typical enhancers by identifying an inflection point of H3K27ac signal; the slope here was 1. Finally, we classified the genes as SE-associated genes by Rose GeneMapper to annotate the genes within the 50-kb range of the super-enhancers.

**Survival analysis of the genes in the cancer genome Atlas dataset**. AML data were used to perform validation with the database (http://gibk21.bse.kyutech.ac.jp/PrognoScan/index.html)[70] and Gene Expression Profiling Interactive Analysis (GEPIA) database (http://gepia.cancer-pku.cn)[71]. The overall survival (OS) was estimated using the log-rank test, and $p$ value < 0.05 was considered to denote statistically significant data.

**Quantitative RT-PCR**. Total RNA was extracted from sorted GFP+ cells using TRIzol reagent (15596026, Invitrogen) and cDNA was synthesized using Prime-Script™ RT Master Mix (RR036A, Takara). Real-time qPCR was performed using SYBR® Premix Ex Taq™ II (RR820A, Takara) on Applied Biosystems™ 7500(Thermo Scientific) system. The exhibited data represents the fold change (FC) of the experimental group versus control group. In brief, ΔCt was calculated as ΔCt = Ct (test gene) - Ct (Ref. gene). ΔΔCt was calculated as ΔΔCt = ΔCt (experimental group) - ΔCt (control group). The FC of a test gene in the experimental group versus control group was calculated as $FC = 2^{(-\Delta\Delta Ct)}$. Each gene tested in triplicates in every independent experiment, and all experiments were triplicated.

**Acute myeloid leukemia mouse model**. The 293 T cells were transfected with retroviral plasmids MSCVMLL-AF9-IRES-GFP containing MLL-AF9 and GFP cDNA sequences. Bone marrow cells from C57 mice treated with 5-fluorouracil (5-FU) for 5 days were infected with retrovirus twice with 24 h interval. The 400 K infected cells were mixed with 100 K protective cells to intravenously inject into WT recipient mice irradiated with a 9-Gy lethal dose. The number of animals used per experiment is shown in the figure legends. *Capg* knockdown and control lentivirus were prepared by HEK293T transfected by pLKO.1-puro together with psPAX2, pMD2.G packaging vectors. MLL-AF9-GFP+ bone marrow cells were harvested from AML mice at 35 days after transplantation. These cells were infected with *Capg* knockdown or control lentivirus and further selected by 1 mg/ml puromycin for 72 h. The 200 K GFP+ cells screened by puromycin were mixed with 100 K protective cells to intravenously inject into CD45.2+ recipient mice irradiated with a 4.5-Gy sublethal dose.

**IP-MS**. To identify CAPG partners in human AML cells, we pulled down CAPG protein complexes from nuclear extraction in THP-1 by CAPG antibody (ab181092, abcam). The THP-1 cells were expanded to 10 of 150 mm diameter dishes for the preparation of total protein. Protein was pre-cleared with 0.5 mL of Protein A + G Agarose beads (P2055, Beyotime) in 16 mL IP DNP buffer(20 mM HEPES, pH 7.9, 20% Glycerol, 100 mM KCl, 1.5 mM MgCl2, 0.2 mM EDTA, 0.02% NP-40, 0.5 mM DTT, 0.2 mM PMSF, 0.1% Protease inhibitor cocktail) overnight at 4 °C. At the same time, 25 μg CAPG antibody or IgG(12-370, Millipore) conjugated Protein G agarose beads by incubating in IP DNP buffer overnight at 4 °C. The pre-cleared nuclear extracts were combined with the antibody-conjugated beads, and rotated for 5 h at 4 °C. After five washes in Buffer D (20 mM HEPES pH = 7.6, 0.2 mM EDTA, 1.5 mM MgCl2, 100 mM KCl, 20% glycerol) supplemented with 0.02% NP-40, the bound material was eluted by boiling for 5 min in 2xSDS loading buffer. Put the solution into the concentration columns, centrifuge $14,000 \times g$ 30 min at 20 °C. Centrifuge until the solution near 40 μl, Samples were then fractionated on Omni-PAGE™Hepes-Tris Gels (LK206, Epizyme) and stained with InstantBlue Protein Stain (AQ211, Analysis Quiz). The products from a single purification were subjected to whole lane LC-MS/MS sequencing and data analysis. Two biological replicates were performed for each antibody.

The raw MS files were analyzed and searched against protein database based on the species of the samples using MaxQuant (1.6.2.10)[72].

**Data analysis**. According to the intensity of each protein in the sample, with the control IgG as the denominator and the CAPG experimental group as the numerator, the fold change value was calculated. We select proteins with FC > 2 as the interactome of CAPG.

The protein interaction and complex data were access directly from the:
STRING Version11.5 (https://cn.string-db.org/)[73],
CORUM Version3.0 (https://mips.helmholtz-muenchen.de/corum/#)[74],
UniProt (https://www.uniprot.org/)[75].

The final confirmed PPI network was assessed by the STRING database, and the recognized individuals were interacted by Cytoscape Version 3.9.1 software.

**ChIP**. 1% formaldehyde in PBS was used to crosslink the cells for 10 min, followed by quenched with 125 mM glycine on ice. Cells were collected and flash-frozen in liquid nitrogen, then stored at −80 °C for use. Frozen crosslinked cells were thawed on ice and then resuspended in lysis buffer I (50 mM HEPES-KOH, pH 7.5, 140 mM NaCl, 1 mM EDTA, 10% glycerol, 0.5% NP-40, 0.25% Triton X-100, protease inhibitors). After rotated for 10 min at 4 °C, the cells were collected, and resuspended in lysis buffer II (10 mM Tris-HCl, pH 8.0, 200 mM NaCl, 1 mM EDTA, 0.5 mM EGTA, protease inhibitors). After rotated for 10 min at 4 °C, the cells were collected, and resuspended in sonication buffer (20 mM Tris-HCl pH 8.0, 150 mM NaCl, 2 mM EDTA pH 8.0, 0.1% SDS, and 1% Triton X-100, protease inhibitors) for sonication. Sonicated lysates were cleared once by centrifugation at $16,000 \times g$ for 10 min at 4 °C. Input material was reserved. The remainder was incubated with magnetic beads bound with CAPG antibody (ab181092, abcam) to enrich for DNA fragments overnight at 4 °C. Beads were washed with wash buffer (50 mM HEPES-KOH pH 7.5, 500 mM LiCl, 1 mM EDTA pH 8.0, 0.7% Na-Deoxycholate, 1% NP-40) and TE buffer (10 mM Tris-HCl pH 8.0, 1 mM EDTA, 50 mM NaCl) in order. Beads were removed by incubation at 65 °C for 30 min in elution buffer (50 mM Tris-HCl pH 8.0, 10 mM EDTA, 1% SDS). Cross-links were reversed overnight at 65 °C. To purify eluted DNA, 200 mL TE was added and then RNA was degraded by incubation in 8 μl 10 mg/mL RNase A at 37 °C for 2 h. Protein was degraded by addition of 4 μl 20 mg/mL proteinase K and incubation at 55 °C for 2 h. Phenol: chloroform: isoamyl alcohol extraction was performed followed by ethanol precipitation. The DNA has then resuspended in 50 ml TE. Library preparation was performed with a DNA Library Prep Kit (Vazyme, #TD501), libraries were amplified for seven cycles, and were size-selected with Beckman AMPure XP beads.

**RNA-seq data analysis**. For read alignment and expression quantification, we first removed low-quality reads, and trimmed the adaptor sequence with Trim Galore. Then we mapped the remaining pair-end reads to the reference genome using STAR with ENCODE option bundles. Using HTSeq-count, we counted the uniquely mapped reads, and normalized the read count by trimmed mean of M values (TMM), and transformed it to reads per kilobases per million reads (RPKM) by edgeR. With an expression cutoff of RPKM R 1 in at least one sample group, we removed low abundant genes, and detected the differentially expressed genes using edgeR. Genes were considered differentially expressed when the overall false discovery rate (FDR) < 0.01 and fold change is above 2.0.

**ChIP-seq analysis**. Fastq files were trimmed adaptors by TrimGalore (version 0.6.4, https://github.com/FelixKrueger/TrimGalore) and aligned to mm9 reference genome using Bowtie2 (version 2.2.5, http://bowtie-bio.sourceforge.net/bowtie2/) with default parameters. Reads with a map score <30 and PCR duplications were filtered out by using Samtools[76] (version 1.9, http://samtools.sourceforge.net). Reads aligned to the regions in ENCODE blacklist (http://mitra.stanford.edu/kundaje/akundaje/release/ blacklists/) were discarded through bedtools[77] (version 2.29.1, https://bedtools.readthedocs.io/en/latest/). Peaks were called with macs2[78] (version 2.1.2, parameters: '-g mm -q 0.05 -m 5 50') using input as control. DiffBind (version 2.10.0) was used to analyze differential binding sites.

**Flow cytometry**. Take 20–30 μL of peripheral blood through the tail vein of the mouse and add it to the anticoagulation tube. Take the bone marrow cells from the femur and tibia of the sacrificed mice. The red blood cells were lysed, and the bone marrow cells were filtered using a 100-mm cell strainer. Monoclonal antibodies to Mac-1 (M1/70, Biolegend), Gr-1 (RB6-8C5, Biolegend), c-Kit (2B8, Biolegend), Lin mix (Gr1, CD4, CD3, CD8a, Ter119, B220, IgM) (Biolegend), CD34 (MEC14.7, Biolegend), Sca1 (D7, Biolegend), FcgRII/III (93, Biolegend), IL-7Ra (A7R34, Biolegend) (all used 50 ng per million cells) were used where indicated. After incubation with antibodies, the samples were analyzed using the Attune NxT flow cytometer (Thermo), and the results were analyzed using FlowJo software. Here, 7-aminoactinomycin D (7-AAD) (A1310, Life Technologies) was used to exclude dead cells.

**Gene ontology analysis**. Enriched ontology terms for CAPG interactome proteins were identified using STRING. GO Biological Process, GO Molecular Function, and GO Cellular Component were referenced to identify ontology terms with the adjusted $p$ value < 0.05.

**Statistics and reproducibility**. Data are expressed as means ± SEM. For all experiments, except the determination of survival, data were analyzed by Student's $t$ tests, and differences were considered statistically significant if $p < 0.05$. The survival of the two groups was analyzed using a log-rank test, and differences were considered statistically significant if $p < 0.05$. *$p < 0.05$, **$p < 0.01$, ***$p < 0.001$.

**Reporting summary**. Further information on research design is available in the Nature Portfolio Reporting Summary linked to this article.

## Data availability
The datasets presented in this study can be found in online repositories. The names of the repository/repositories and accession number can be found below: NCBI BioProject PRJNA876046. Antibody and oligo information can be found in Supplementary Data 1 and 2. Source data for the IP-MS are provided as Supplementary Data 3. CAPG KD RNA-seq data can be found in Supplementary Data 4. Uncropped scans of the blots were shown in Supplementary Fig. 7. Additional data supporting the findings of this study are available from the corresponding author upon request.

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

## Acknowledgements

We would like to thank the Peking University Medicine Fund of Fostering Young Scholars' Scientific & Technological Innovation (BMU2022PYB014) supported by "the Fundamental Research Funds for the Central Universities"; Scientific Research Seed Fund of Peking University First Hospital (2022SF09); National Natural Science Foundation of China (NSFC 81870127) (NSFC 82072369) (NSFC 82200178).

## Author contributions

Q.M. designed and performed most of the experiments and analyzed the data. M.Z. and B.L. analyzed the data. H.L. supervised the project. All authors contributed to the article and approved the submitted version.

## Competing interests

The authors declare no competing interests.

## Ethics approval and consent participate

Sample collection was approved from the institutional review board and ethics committee of Peking University First Hospital.
