## [Peer Review File · Communications Biology]

Reviewers' comments:

Reviewer #1 (Remarks to the Author):

The following manuscript by the Li group sought to identify new super enhancers (SE) in AML as potential new therapeutic targets. Despite an exciting era of new molecular therapies emerging in AML, this remains a heterogeneous and devastating disease and better understanding of how it arises is required to increase/improve the range and quality of therapeutics that can be deployed. Through this objective the authors identified an interesting new target, CAPG, which appears to have a role in promoting leukaemogenic processes in AML. Additionally the work provides insight into how a cytoskeletal component can affect gene expression and cell signalling activity. Consequently, the manuscript is timely and novel, in addition to being clearly presented and well written.

The work presented will be appealing to the broad readership of this journal, however there are some minor and major issues that require addressing ahead of its publication. My recommendations are below:

Major:

1. In Figure 4e, the authors claim reduction of NFkB targets in response to CAPG knockdown in THP1 cells via shRNA. Confirmation of CAPG KD in THP1 cells is only confirmed at the transcript level and needs confirmation at the protein level given the long protein half life's of many cytoskeletal components. Figure 5b from the in vivo model also requires a protein confirmation of CAPG KD if enough cellular material can be extracted.
2. Also in Figure 4e, NFkB targets are presented which are confirmed targets in other cell types. Confirmation that these are also NFkB targets in THP-1 cells by doing some kind of NFkB interference and reanalysis of gene expression would be helpful in confirming CAPG is impacting bona fide target genes of the pathway.
3. The authors conclude between lines 209-213 that CAPG can cause AML progression through the NFkB pathway. Its clear from the data that CAPG can impact AML cells but the authors provide no evidence that this is via NFkB signalling and so this does not justify the conclusion. The authors should either remove/soften this part or execute some kind of rescue experiment by re-activating the NFkB pathway in CAPG KD cells showing that it perturbs any anti-leukaemia phenotype observed. Alternatively this could be done in cell lines as a proof of principle.

Minor:

1. The opening line of the introduction (lines 44-46) makes little sense: Acute myeloid leukemia (AML) is an aggressive hematologic malignancy that is mainly caused by the early stages of hematopoietic stem cell development and malignant proliferation.

AML is not caused by HSC development, but rather emerges from genetic mutations which occur in this cell type.

2. Lines 152-166 are really descriptive for a results section and read more like a discussion. This should either be restructured to read more like a results section or be moved to the discussion section.
3. Figure 5c, the authors present data showing a reduction in %GFP+ cell in the bone marrow of shCAPG mice versus controls. If available, a further breakdown of immunophenotyping would be

helpful here to show that is specifically cells with a myeloid phenotype that are affected here and this relevant to an AML theme. The inclusion of murine myeloid/stem markers such as Sca1, Kit, Gr1 or Mac1 would be very informative and would rule out the possibility that any affect being observed is arising from off-target effects on the lymphoid compartment.

Reviewer #2 (Remarks to the Author):

The overall objective of the manuscript is to identify super enhancers and confirm a role in a acute myeloid leukemia (AML). The authors identify CAPG and provide shRNA Knock down studies as well as MLL murine studies to confirm a role in AML development. Specifically, the authors attempt to relate this role to NF- κ B but little experimental evidence is provided regarding this aspect. Further, there are several areas of the manuscript that can be improved to improve readability and clarity. In addition to additional experiments, the figure legends are poor and lack significant detail (though this is provided in methods).

Significant issue:

Figure 1 – It is not entirely clear how each of these analyses were performed or where the individual data points come from. I think Supplementary data for each of those individual approaches to yield the data for the 'integrated analysis' needs to be shown. The legends of the figures need to be improved and interpreting the figures without the methods is very difficult. For example, I had to find within the text what NE/MO referred to, as opposed to being part of the legend for Fig 1A. Purple is a poor choice of colour (very similar to black and so that part of the Venn diagram is not immediately). Why is the intersection '6' chosen -i can guess but the rationale is not clear in the text. The experimental scheme and description therefore needs significant improvements to improve clarity.

S.Figure 1. Again poor legend description. Specifically, what data was used, were all patients included? What are the normals? Depending on data sets used, TCGA have hundreds of patients (like Fig 1d). The data points don't reflect that and its not clear why?. In SFig2B, what is the model used in part b, what transplants experiments -at least some indication here in the legend would be helpful.

Figure 2 and associated text. 'highly expressed' – what does this mean? Is the data statistically significant? Kaplan-M. curves are not all defined in the legend (e.g. what are the gray dotted lines? Similar queries for SF2). How is high and low TPM defined? In Figure 2d, the controls are not defined (white box and whiskers I assume?)

Line 122- There has been some published data on CAPG in AML. It is surprising that this manuscript has not identified and discussed:- Pierce, A. et al. Eight-channel iTRAQ enables comparison of the activity of six leukemogenic tyrosine kinases. *Mol Cell Proteomics* 7, 853-863, doi:10.1074/mcp.M700251-MCP200 (2008). Also this article in ALL: Verrills, N. M. et al. Proteomic analysis reveals a novel role for the actin cytoskeleton in vincristine resistant childhood leukemia--an in vivo study. *Proteomics* 6, 1681-1694, doi:10.1002/pmic.200500417 (2006).

Figure 3 – So far data has been transcriptomic and there is little information regarding protein expression of CAPG in normal versus AML in human or mice. Are the changes also observed at the protein level? This is also important to show and validate the knock down approach in THP1 cells. The authors need to provide evidence of the proteomic approach, pull down etc. Figure 3B – do the colours indicate anything. What programs/software was used for the interactome analysis?

Figure 4- As above. no evidence is provided for demonstraig KD has been successful in THP1 cells at

the protein level; relative expression (assuming qPCR or mRNA) is shown. Also it is not clear if only 1 shRNA has been used. The experiments need to be confirmed with at least 1 more shRNA to rule out non specific effects. This has been done in Supp Figure 4.

SF4. Only 1 patient is shown. Can this be repeated in more than 1 AML patient? What are the clinical characteristics of these patients. Further only 1 cell line is tested. What happens if you KD in cell lines without capg? Hopefully no effect.

Figure 5b. Again only mRNA transcripts of capg are shown to validate KD. However, this does not demonstrate that KD has been achieved and how much KD of protein is achieved.

Reviewer #3 (Remarks to the Author):

The manuscript entitled "Super-enhancer-associated gene CAPG promotes AML progression " by Ma and Li describe the role of CAPG in AML. The gelsolin-like actin-capping protein (CAPG) is an actin-binding protein in the gelsolin superfamily. Multiple studies have reported that CAPG is highly expressed in various types of cancer. However, the role of CAPG in malignant tumors and its prognostic significance is still controversial. Nonetheless, current genomic and proteomic analysis provides compelling evidence for CAPG's role in tumor maintenance and progression. The manuscript is well written and supports the finding of Nf-kB activation in AML progression.

Major Comments:

1. Role of superenhancers in cancer and more specifically in AML is well described. Mapping of superenhancers and their comparisons between normal and AML samples are not properly described. Also, it is not clear why superenhancer analysis is more revealing than simply doing the gene expression analysis.
2. Author's noted induction of six SE associated genes (CAPG, CD207, GPR132, SLC7A11, HIPK3 and FCER1G), it is not clear why they focused on only CAPG. Are these genes have been validated and characterized? If so, please cite and discuss.
3. Given the genetic heterogeneity in AML specifically in accumulating mutations in epigenetic regulating genes, it is necessary to determine the expression and mapping the SE in other AML subtypes. For instance, mouse or human AML cells with mutations harboring in genes FLT3ITD, TET2, DNMT3A, P53, RAS, and IDH1/2 should be evaluated.
4. It is not clear whether Capg down regulation affects global transcriptional machinery, or it specifically alters the NfKb pathway genes. Analysis of cytoplasmic and nuclear cytoskeleton should be performed to determine whether CAPG affects nuclear matrix or directly regulates NFkB genes.
5. A pull down and western blotting of interacting proteins (NSL complex, SNW1 complex, MLL1-WDR5 complex,) is important for validation.
6. Likewise, pulldown and western blotting of CCAR2, RPL4 and ZFP91 is essential to establish the interaction.
7. CAPG chip-seq data is not properly described. It is not clear how many peaks were determined and what control cells were employed. How these peaks are different in normal and leukemic cells. Perhaps including CAPG KD/KO cells in CHIP-seq data gathering and analysis may provide more clarity.
8. Given the deregulation of actin remodeling is one of central feature of most cancers, actin KD in MLL-AF9 leukemia should be assessed and compared with CAPG.

Minor points.

1. Actin polymerization and cellular cytoskeleton should be determined in leukemic cells.
2. Please provide evidence and discuss why analysis of SE genes are more effective in mitigating the disease than other OE genes.

Author response to Reviewers' comments

Reviewer #1:

The following manuscript by the Li group sought to identify new super enhancers (SE) in AML as potential new therapeutic targets. Despite an exciting era of new molecular therapies emerging in AML, this remains a heterogeneous and devastating disease and better understanding of how it arises is required to increase/improve the range and quality of therapeutics that can be deployed. Through this objective the authors identified an interesting new target, CAPG, which appears to have a role in promoting leukaemogenic processes in AML. Additionally the work provides insight into how a cytoskeletal component can affect gene expression and cell signalling activity. Consequently, the manuscript is timely and novel, in addition to being clearly presented and well written.

The work presented will be appealing to the broad readership of this journal, however there are some minor and major issues that require addressing ahead of its publication. My recommendations are below:

Thank you for your valuable feedback. I appreciate your time and effort in reviewing my work. I have carefully considered your comments and made the necessary revisions in the main text.

Major:

1. In Figure 4e, the authors claim reduction of NFkB targets in response to CAPG knockdown in THP1 cells via shRNA. Confirmation of CAPG KD in THP1 cells is only confirmed at the transcript level and needs confirmation at the protein level given the long protein half life's of many cytoskeletal components. Figure 5b from the in vivo model also requires a protein confirmation of CAPG KD if enough cellular material can be extracted.

Response:

We thank the reviewer for the key question about protein level. We collected THP-1 cells, and AML cells from AML mice, checked the protein expression of CAPG in the control and CAPG KD group (Figure 5A, 5B).

Figure 5

(A) Western blot analysis showing CAPG knockdown in THP-1 cells.

(B) Western blot analysis showing Capg knockdown in sorted AML cells from AML mice bone marrow.

2. Also in Figure 4e, NFkB targets are presented which are confirmed targets in other cell types. Confirmation that these are also NFkB targets in THP-1 cells by doing some kind of NFkB interference and reanalysis of gene expression would be helpful in confirming CAPG is impacting bona fide target genes of the pathway.

Response:

Thank you for your inquiry. In order to address this question, we first conducted a ChIP-qPCR experiment for *CAPG* in THP-1 cells and validated that *CAPG* targets the transcription factors of NF-κB pathway (Figure 6A). We then collected RNA-seq data from cells with *CAPG* knockdown, which revealed that the down-regulated genes after *CAPG* knockdown were significantly enriched on the NF-κB pathway (Figure 6B, 6C). These results collectively suggest that *CAPG* impact NF-κB pathway in THP-1 cells.

We have made the corresponding revisions on lines 173-176 of the manuscript.

Figure 6

(A) The binding fold change of CAPG at the NF-κB pathway transcription factor family by ChIP-qPCR. Data are presented as means ± SD. *p < 0.05, **p < 0.01. Three biological replicates are assayed for ChIP-qPCR experiment.

(B) RNA-seq data analysis the expression level in THP-1 cells after *CAPG* knockdown.

(C) KEGG analysis show the down-regulated genes significant enriched on the NF-κB pathway.

3. The authors conclude between lines 209-213 that CAPG can cause AML progression through the NFkB pathway. It's clear from the data that CAPG can impact AML cells but the authors provide no evidence that this is via NFkB signalling and so this does not justify the conclusion. The authors

should either remove/soften this part or execute some kind of rescue experiment by re-activating the NFkB pathway in CAPG KD cells showing that it perturbs any anti-leukaemia phenotype observed. Alternatively this could be done in cell lines as a proof of principle.

Response:

Thank you for your question. To demonstrate the relationship between CAPG and NF- κ B, we conducted a rescue experiment. TNF- α is an activator of the NF- κ B pathway(Cheng et al., 2021).

After knocking down CAPG, we added TNF- α to rescue the effects on the NF- κ B pathway downstream genes (Figure 7A). *CCL5*, *TNFRSF13C*, *CCR5*, *TLR2*, *BCL2*, *TRAF1*, *EGR1*, *IER3* are all downstream genes of the NF- κ B pathway, which are associated with immune response and inflammation. We observed a decrease in leukemogenesis expression after knocking down CAPG. We activated the pathway by stimulating THP-1 cells with TNF- α and observed a significant rescue of downstream genes expression (Figure 7B). These results provide supportive evidence for the conclusion that *CAPG* can modify AML progression through NF- κ B.

MYC, *KIT*, *FLT3L*, *IKZF2*, *SOX4* are well-known genes involved in leukemogenesis. When *CAPG* is knocked down, these genes are significantly down-regulated, suggesting that *CAPG* may be able to resist the progression of leukemia (Figure 7C).

We have made the corresponding revisions on lines 181-189 of the manuscript.

- (A) mRNA and protein levels of CAPG in control, knockdown, and 10ng/mL TNF- α stimulated group were compared. Data are presented as means \pm SD. ***p < 0.001.
- (B) The addition of TNF- α reversed the inhibition of NF- κ B pathway activity induced by CAPG knockdown in THP-1 cells. Data are presented as means \pm SD. **p < 0.01, ***p < 0.001.
- (C) mRNA levels of leukemogenesis in control, CAPG knockdown group. Data are presented as means \pm SD. **p < 0.01, ***p < 0.001.

Minor:

1. The opening line of the introduction (lines 44-46) makes little sense: Acute myeloid leukemia (AML) is an aggressive hematologic malignancy that is mainly caused by the early stages of hematopoietic stem cell development and malignant proliferation. AML is not caused by HSC development, but rather emerges from genetic mutations which occur in this cell type.

Response:

Thank you for your valuable feedback. I have carefully considered your comments and made the revision in the manuscript.

We have made the corresponding revisions on lines 45-47 of the manuscript.

2. Lines 152-166 are really descriptive for a results section and read more like a discussion. This should either be restructured to read more like a results section or be moved to the discussion section.

Response:

Thank you for your valuable feedback. I have carefully considered your comments and made the revision in the manuscript.

We have made the corresponding revisions on lines 278-286 of the manuscript.

3. Figure 5c, the authors present data showing a reduction in %GFP+ cell in the bone marrow of shCAPG mice versus controls. If available, a further breakdown of immunophenotyping would be helpful here to show that is specifically cells with a myeloid phenotype that are affected here and this relevant to an AML theme. The inclusion of murine myeloid/stem markers such as Sca1, Kit, Gr1 or Mac1 would be very informative and would rule out the possibility that any affect being observed is arising from off-target effects on the lymphoid compartment.

Response:

Thank you for your question. In the Methods section of manuscript, we mentioned the use of Mac-1, Gr-1, c-Kit, Lin mix (Gr1, CD4, CD3, CD8a,

Ter119, B220, IgM), CD34, Sca1, FcγRII/III, IL-7Ra antibody to eliminate interference. Previous publications have shown that in this type of mouse model (Somerville and Cleary, 2006), the majority of cells are Gr1⁺ and Mac⁺, with only a small proportion of other cells. Therefore, the decrease in the percentage of %GFP⁺ cells is caused by a reduction in *CAPG* expression.

There is a corresponding description on lines 433-440 of the manuscript.

Reviewer #2 (Remarks to the Author):

The overall objective of the manuscript is to identify super enhancers and confirm a role in acute myeloid leukemia (AML). The authors identify CAPG and provide shRNA Knock down studies as well as MLL murine studies to confirm a role in AML development. Specifically, the authors attempt to relate this role to NF- κ B but little experimental evidence is provided regarding this aspect. Further, there are several areas of the manuscript that can be improved to improve readability and clarity. In addition to additional experiments, the figure legends are poor and lack significant detail (though this is provided in methods).

We would like to thank the reviewer for the comments and the suggestions to improve the manuscript. We answer the questions one by one, and the details are shown below:

Significant issue:

Figure 1 – It is not entirely clear how each of these analyses were performed or where the individual data points come from. I think Supplementary data for each of those individual approaches to yield the data for the ‘integrated analysis’ needs to be shown. The legends of the figures need to be improved and interpreting the figures without the methods is very difficult. For example, I had to find within the text what NE/MO referred to, as opposed to being part of the legend for Fig 1A. Purple is a poor choice of colour (very similar to black and so that part of the Venn diagram is not immediately). Why is the intersection ‘6’ chosen - I can guess but the rationale is not clear in the text. The experimental scheme and description therefore needs significant improvements to improve clarity.

Response:

Thank you for your attention to detail, your feedback has been very helpful in improving my manuscript. In response to your questions and suggestions, we have made the following revisions:

1. We have added a separate SE analysis section in the article and uploaded detailed analysis results in the supplementary materials (Figure 8A). We have also provided a more detailed description of the analysis in the figure legend.
2. We have made corrections to the manuscript by including the full names of NE (neutrophils), MO (monocytes). We have made the corresponding revisions on lines 675-676 of the manuscript.
3. We have adjusted the color scheme of the images to make them more suitable for publication. We have made the corresponding revisions on figure 1A of the manuscript.

4. We have supplemented a detailed introduction to the candidate genes as requested by the reviewer. We have made the corresponding revisions on lines 234-251 of the manuscript.

Figure 8

(A) Enhancers in three normal blood cells and three AML cells ranked based on H3K27ac signal intensity.

S.Figure 1. Again poor legend description. Specifically, what data was used, were all patients included? What are the normals? Depending on data sets used, TCGA have hundreds of patients (like Fig 1d). The data points don't reflect that and its not clear why?. In S.Fig2B, what is the model used in part b, what transplants experiments -at least some indication here in the legend would be helpful.

Response:

Thank you for your inquiry.

1. We would like to confirm that S.Figure 1 employs RNA-seq data from GSE128910, which comprises a total of 7 samples with each data point representing one individual. Of the 7 samples, 4 were derived from healthy individuals and 3 were derived from individuals with AML patients.
2. Additionally, S.Figure 2 showcases the MLL-AF9 mouse model cells that were utilized in our study. We have made sure to address these modifications in the manuscript as well.

We have made the corresponding revisions on the supplement figure of the manuscript.

Figure 2 and associated text. 'highly expressed' – what does this mean? Is the data statistically significant? Kaplan-M. curves are not all defined in the legend (e.g. what are the gray dotted lines? Similar queries for SF2). How is high and low TPM defined? In Figure 2d, the controls are not defined (white box and whiskers I assume?)

Response:

Thank you for your inquiry. I will address these questions one by one and make the necessary corrections in the manuscript.

1. The mouse model data statistically significantly highly expressed, $P=0.0147$.
2. Dot lines represent 95% Confidence bound. TPM stands for "transcripts per million" and is a measure of gene expression levels. High TPM could be defined as expression levels above the median, and low TPM could be defined as expression levels below the median. We have made the corresponding revisions on line688 of the manuscript.
3. White box is the control group that I have indicated in the image.

Line 122- There has been some published data on CAPG in AML. It is surprising that this manuscript has not identified and discussed: - Pierce, A. et al. Eight-channel iTRAQ enables comparison of the activity of six leukemogenic tyrosine kinases. *Mol Cell Proteomics* 7, 853-863, doi:10.1074/mcp.M700251-MCP200 (2008). Also this article in ALL: Verrills, N. M. et al. Proteomic analysis reveals a novel role for the actin cytoskeleton in vincristine resistant childhood leukemia--an in vivo study. *Proteomics* 6, 1681-1694, doi:10.1002/pmic.200500417 (2006).

Response:

Thank you for your careful review and corrections. We have now incorporated the section you highlighted into the manuscript, which has made my article clearer and more precise. We have made the corresponding revisions on lines 244-246 of the manuscript.

Figure 3 – So far data has been transcriptomic and there is little information regarding protein expression of CAPG in normal versus AML in human or mice. Are the changes also observed at the protein level? This is also important to show and validate the knock down approach in THP1 cells. The authors need to provide evidence of the proteomic approach, pull down etc. Figure 3B – do the colours indicate anything. What programs/software was used for the interactome analysis?

Response:

Thank you for your question. We collected leukemic cells from primary transplant mice compared with normal mouse bone marrow cells, checked the protein expression of CAPG (Figure 9A). We have included protein level experiments following knockdown of the CAPG in both THP-1 cells and mouse model (Figure 9B, 9C). Additionally, we have conducted pull-down experiments to validate the results of IP-MS analysis (Figure 9D, 9E). When presenting the protein interaction network results, we utilized the STRING (Version11.5 <https://cn.string-db.org/>) and Cytoscape software (Version 3.9.1 software) to analyze and visualize the protein-protein interaction network. The different colors in the network represent different functional categories.

Figure 9

(A) Western blot showing that CAPG expression was increased in leukemic cells from primary transplant mice compared with normal mouse bone marrow cells. GAPDH was used to show equal loading. Each band represent an individual mouse.

(B) Western blot analysis showing CAPG knockdown in THP-1 cells.

(C) Western blot analysis showing Capg knockdown in sorted AML cells from AML mice bone marrow.

(D) Co-IP shows the interactions between CAPG and NF- κ B related proteins (RPL4, ZFP91, CCAR2).

(E) Co-IP shows the interactions between CAPG and NSL complex (WDR5, MCRS1), SNW1 complex (EEFG), MLL-WDR5 complex (KMT2A, WDR5).

Figure 4- As above. no evidence is provided for demonstraign KD has been successful in THP1 cells at the protein level; relative expression (assuming qPCR or mRNA) is shown. Also it is not clear if only 1 shRNA has been used. The experiments need to be confirmed with at least 1 more shRNA to rule out non specific effects. This has been done in Supp Figure 4.

Response:

We thank the reviewer for question. We designed three shRNAs, but shRNA3# did not work as expected (Figure 10B). Therefore, we only presented the results of two shRNAs in Supplementary Figure 6. We also selected the most efficient one for subsequent experiments. We also collected THP-1 cells, checked the protein expression of CAPG in the control and CAPG KD group (Figure 10B).We selected the shRNA with the highest efficiency for experimentation based on the results from RT-qPCR.

Figure 10

(A) The mRNA expression level of *CAPG* after knockdown with different shRNAs.

(B) Western blot analysis showing *CAPG* knockdown in THP-1 cells.

SF4. Only 1 patient is shown. Can this be repeated in more than 1 AML patient? What are the clinical characteristics of these patients. Further only 1 cell line is tested. What happens if you KD in cell lines without capg? Hopefully no effect.

Response:

Thank you for your question. THP-1 and Molm-13 are both human-derived cell lines. The THP-1 cell line is a human monocytic leukemia cell line that was originally isolated from the peripheral blood of a 1-year-old male with acute monocytic leukemia in 1980(Tsuchiya et al., 1980). The MOLM-13 cell line were established from the peripheral blood of a 20-year-old male with AML-M5a(Matsuo et al., 1997). The RNA-seq data we used also comes from patients. *CAPG* is highly expressed in these cells.

Figure 5b. Again, only mRNA transcripts of capg are shown to validate KD. However, this does not demonstrate that KD has been achieved and how much KD of protein is achieved.

Response:

We thank the reviewer for the key question about protein level. We collected THP-1 cells (Figure 11A), and AML cells from AML mice (Figure 11B), checked the protein expression of CAPG in the control and *CAPG* KD group.

Figure 11

(A) Western blot analysis showing *CAPG* knockdown in THP-1 cells.

(B) Western blot analysis showing *Capg* knockdown in sorted AML cells from AML mice bone marrow.

Reviewer #3 (Remarks to the Author):

The manuscript entitled “Super-enhancer-associated gene CAPG promotes AML progression” by Ma and Li describe the role of CAPG in AML. The gelsolin-like actin-capping protein (CAPG) is an actin-binding protein in the gelsolin superfamily. Multiple studies have reported that CAPG is highly expressed in various types of cancer. However, the role of CAPG in malignant tumors and its prognostic significance is still controversial. Nonetheless, current genomic and proteomic analysis provides compelling evidence for CAPG’s role in tumor maintenance and progression. The manuscript is well written and supports the finding of Nf-kB activation in AML progression.

Thank you for your valuable feedback. I appreciate your time and effort in reviewing my work. I have carefully considered your comments and made the necessary revisions in the main text.

Major Comments:

1. Role of superenhancers in cancer and more specifically in AML is well described. Mapping of superenhancers and their comparisons between normal and AML samples are not properly described. Also, it is not clear why superenhancer analysis is more revealing than simply doing the gene expression analysis.

Response:

Thank you for your inquiry. We have provided a detailed description of the super-enhancer data for normal and AML samples in the Supplement Figure 1. RNA-seq analysis at the transcriptome level can generate a large number of differentially expressed genes, making it difficult to identify genes that significantly impact disease progression. In contrast, super-enhancers are highly cell-type specific and have powerful gene expression regulatory functions, which can drive the expression of genes that control and define cell identity (Hnisz et al., 2013). By integrating super-enhancer data with transcriptomic data, we can narrow the target range and improve accuracy. I have made the necessary changes to the discussion part.

We have made the corresponding revisions on lines 225-231 of the manuscript.

2. Author’s noted induction of six SE associated genes (CAPG, CD207, GPR132, SLC7A11, HIPK3 and FCER1G), it is not clear why they focused on only CAPG. Are these genes have been validated and characterized? If so, please cite and discuss.

Response:

Thank you for pointing out the issues, I have made the necessary changes to the discussion part. We have made the corresponding revisions on lines 234-246 of the manuscript.

3. Given the genetic heterogeneity in AML specifically in accumulating mutations in epigenetic regulating genes, it is necessary to determine the expression and mapping the SE in other AML sub-types. For instance, mouse or human AML cells with mutations harboring in genes FLT3ITD, TET2, DNMT3A, P53, RAS, and IDH1/2 should be evaluated.

Response:

Thank you for your question. EZH2 is a histone methyltransferase. We analyzed the H3K27ac ChIP-seq data from leukemia cell groups with and EZH2 knockout group(Basheer et al., 2019), and identified super-enhancers. After knocking out EZH2, there was no significant change in the H3K27ac peaks or super-enhancer of the *Capg* region (Figure 12A).

Figure 12

(A) The ChIP-seq tracks show the representative H3K27ac signal in WT and EZH2 KO AML cells. The super-enhancers are shown as blue or red boxes.

4. It is not clear whether *Capg* down regulation affects global transcriptional machinery, or it specifically alters the NfKb pathway genes. Analysis of cytoplasmic and nuclear cytoskeleton should be performed to determine whether CAPG affects nuclear matrix or directly regulates NFkB genes.

Response:

Thank you for your question. We analyzed the expression levels of the actin and other cell cytoskeleton-related genes after downregulating *CAPG* through RNA-seq data (Figure 13C), and there was no significant change in the expression trend (Figure 13A,13B). This suggests that *CAPG* regulates NF- κ B transcription factors not through cytoskeleton proteins.

Figure 13

- (A) Changes in actin and cytoskeleton-related genes following CAPG knockdown analyzed by Gene Set Enrichment Analysis (GSEA).
- (B) Volcano plot showing changes in expression levels of actin and cytoskeleton-related genes following CAPG knockdown.
- (C) Gene list of various components of cytoskeleton-related genes.

5. A pull down and western blotting of interacting proteins (NSL complex, SNW1 complex, MLL1- WDR5 complex,) is important for validation.

Response:

Thank you for your question. We have conducted pull-down experiments to validate the results of IP-MS analysis (Figure 14A).

Figure 14

- (A) Co-IP shows the interactions between CAPG and NSL complex (WDR5, MCRS1), SNW1 complex (EEF1G), MLL-WDR5 complex (KMT2A, WDR5).

6. Likewise, pulldown and western blotting of CCAR2, RPL4 and ZFP91 is essential to establish the interaction.

Response:

Thank you for your question. We have conducted pull-down experiments to validate the results of IP-MS analysis (Figure 15A).

Figure 15

(A) Co-IP shows the interactions between CAPG and NF- κ B related proteins (RPL4, ZFP91, CCAR2).

7. CAPG chip-seq data is not properly described. It is not clear how many peaks were determined and what control cells were employed. How these peaks are different in normal and leukemic cells. Perhaps including CAPG KD/KO cells in CHIP-seq data gathering and analysis may provide more clarity.

Response:

Thank you for your question. We used normal C57BL mice as the control group and AML model mice as the experimental group, and analyzed ChIP-seq data to identify peaks. Because CAPG is a cytoskeletal protein, there was no significant difference in the number of peaks between the two groups (Figure 16A).

The expression level of CAPG in normal mouse bone marrow cells is lower than that in leukemia cells. Therefore, we used normal mouse bone marrow cells as a reference to detect the peaks of CAPG enrichment on the genome. After comparison, we found that AML-specific peaks were significantly enriched in the region of NF- κ B transcription factors (Figure 16B).

Figure 16

- (A) Venn diagram of the CAPG binding loci in normal bone marrow cells and AML bone marrow cells.
- (B) Genome browser views of the distribution of CAPG ChIP-seq peaks on NF-κB related genes loci.

8. Given the deregulation of actin remodeling is one of central feature of most cancers, actin KD in MLL-AF9 leukemia should be assessed and compared with CAPG.

Response:

Thank you for your question. We found that knocking down *CAPG* did not significantly decrease the expression of *ACTB* (Figure 17A). CCK-8 experiments showed that knocking down both *CAPG* or *ACTB* resulted in a significant decrease in cell proliferation (Figure 17B). While indicating that *ACTB* may also affect AML, the impact of *CAPG* may be unrelated to *ACTB*.

Figure 17

- (A) The relative mRNA expression level of *CAPG* and *ACTB* following knockdown experiment.
- (B) Cell proliferation was evaluated by cell counting kit 8 cell viability assay following knockdown *CAPG* or *ACTB*.

Minor points.

1. Actin polymerization and cellular cytoskeleton should be determined in leukemic cells.

Response:

Thank you for your question. We analyzed RNA-seq data from the health control group and AML patient group and found that genes related to actin polymerization and cellular cytoskeleton are relatively upregulated in AML.

Figure 18

- (A) Compare the expression level of cellular cytoskeleton related genes from health control group and AML patient group.

2. Please provide evidence and discuss why analysis of SE genes are more effective in mitigating the disease than other OE genes.

Response:

RNA-seq analysis at the transcriptome level can generate a large number of differentially expressed genes, making it difficult to identify genes that significantly impact disease progression. In contrast, super-enhancers are highly cell-type specific and have powerful gene expression regulatory functions, which can drive the expression of genes that control and define cell identity (Hnisz et al., 2013). By integrating super-enhancer data with transcriptomic data, we can narrow the target range and improve accuracy.

Basheer, F., Giotopoulos, G., Meduri, E., Yun, H., Mazan, M., Sasca, D., Gallipoli, P., Marando, L., Gozdecka, M., Asby, R., *et al.* (2019). Contrasting requirements during disease evolution identify EZH2 as a therapeutic target in AML. *J Exp Med* *216*, 966-981.

Cheng, Q.J., Ohta, S., Sheu, K.M., Spreafico, R., Adelaja, A., Taylor, B., and Hoffmann, A. (2021). NF-kappaB dynamics determine the stimulus specificity of epigenomic reprogramming in macrophages. *Science* *372*, 1349-1353.

Hnisz, D., Abraham, B.J., Lee, T.I., Lau, A., Saint-Andre, V., Sigova, A.A., Hoke, H.A., and Young, R.A. (2013). Super-enhancers in the control of cell identity and disease. *Cell* *155*, 934-947.

Matsuo, Y., MacLeod, R.A., Uphoff, C.C., Drexler, H.G., Nishizaki, C., Katayama, Y., Kimura, G., Fujii, N., Omoto, E., Harada, M., *et al.* (1997). Two acute monocytic leukemia (AML-M5a) cell lines (MOLM-13 and MOLM-14) with interclonal phenotypic heterogeneity showing MLL-AF9 fusion resulting from an occult chromosome insertion, ins(11;9)(q23;p22p23). *Leukemia* *11*, 1469-1477.

Somervaille, T.C., and Cleary, M.L. (2006). Identification and characterization of leukemia stem cells in murine MLL-AF9 acute myeloid leukemia. *Cancer Cell* *10*, 257-268.

Tsuchiya, S., Yamabe, M., Yamaguchi, Y., Kobayashi, Y., Konno, T., and Tada, K. (1980). Establishment and characterization of a human acute monocytic leukemia cell line (THP-1). *Int J Cancer* *26*, 171-176.

REVIEWERS' COMMENTS:

Reviewer #1 (Remarks to the Author):

The authors have adequately addressed all of my concerns and I congratulate them all on an insightful and interesting piece of work.

Reviewer #2 (Remarks to the Author):

Comments addressed

Reviewer #3 (Remarks to the Author):

Authors have successfully addressed the raised concerns. I support its publication.